# Investigating the Role of Insight, Decision-Making and Mentalizing in Functional Outcome in Schizophrenia: A Cross-Sectional Study

**DOI:** 10.3390/bs12020028

**Published:** 2022-01-27

**Authors:** Paula Jhoana Escobedo-Aedo, Ana Forjan-González, Adela Sánchez-Escribano Martínez, Verónica González Ruiz-Ruano, Sergio Sánchez-Alonso, Laura Mata-Iturralde, Laura Muñoz-Lorenzo, Enrique Baca-García, Anthony S. David, Javier-David Lopez-Morinigo

**Affiliations:** 1Departamento de Psiquiatría, IIS-Fundación Jiménez Díaz, 28040 Madrid, Spain; paula.escobedo@quironsalud.es (P.J.E.-A.); aforjang@gmail.com (A.F.-G.); adela.sancheze@quironsalud.es (A.S.-E.M.); veronicagrr@gmail.com (V.G.R.-R.); SSanchezA@fjd.es (S.S.-A.); mataiturralde@yahoo.es (L.M.-I.); lauramlorenzo@hotmail.com (L.M.-L.); ebacgar2@yahoo.es (E.B.-G.); 2Departamento de Psiquiatría, Universidad Autónoma de Madrid, 28029 Madrid, Spain; 3Investigación Biomédica en Red de Salud Mental, CIBERSAM, 28007 Madrid, Spain; 4Psychology Department, Universidad Católica del Maule, Talca 3460000, Chile; 5Division of Psychiatry, Faculty of Brain Sciences, Institute of Mental Health, University College London, London WC1E 6BT, UK; anthony.s.david@ucl.ac.uk; 6Department of Child and Adolescent Psychiatry, Institute of Psychiatry and Mental Health, Hospital General Universitario Gregorio Marañón, IiSGM, CIBERSAM, School of Medicine, Universidad Complutense, 28009 Madrid, Spain

**Keywords:** metacognition, schizophrenia spectrum disorders, disability, functioning, quality of life

## Abstract

Background: Recovery has become a priority in schizophrenia spectrum disorders (SSDs). This study aimed to investigate predictors of objective—general functioning and disability—and subjective—quality of life (QoL)—measures of functional outcomes in SSD. Methods: Sample: *n* = 77 SSD outpatients (age 18–64, IQ > 70) participating in a randomised controlled trial. Baseline data were used to build three multivariable linear regression models on: (i) general functioning—General Assessment of Functioning (GAF); (ii) disability—the World Health Organization Disability Assessment Schedule (WHODAS-2.0); and (iii) QoL—Satisfaction Life Domains Scale (SLDS). Results: Young age and being employed (R^2^ change = 0.211; *p* = 0.001), late adolescence premorbid adjustment (R^2^ change = 0.049; *p* = 0.0050), negative symptoms and disorganization (R^2^ change = 0.087; *p* = 0.025) and Theory of Mind (R^2^ change = 0.066, *p* = 0.053) predicted general functioning. Previous suicidal behaviour (R^2^ change = 0.068; *p* = 0.023) and negative and depressive symptoms (R^2^ change = 0.167; *p* = 0.001) were linked with disability. Previous suicidal behaviour (R^2^ change = 0.070, *p* = 0.026), depressive symptoms (R^2^ change = 0.157; *p* < 0.001) and illness recognition (R^2^ change = 0.046, *p* = 0.044) predicted QoL. Conclusions: Negative, disorganization and depressive symptoms, older age, unemployment, poor premorbid adjustment, previous suicide attempts and illness awareness appear to underlie a poor global functional outcome in SSD. Achieving recovery in SSD appears to require both symptomatic remission (e.g., through antipsychotics) and measures to improve mastery and relieve low mood.

## 1. Introduction

Schizophrenia spectrum disorders (SSDs) are highly frequent across the world. In particular, incidence of SSD has been recently estimated at approximately 21.4/100,000 person–years in Europe [1]. SSDs remain associated with significant disability, worse quality of life (QoL) and higher mortality rates compared with the general population [2].

Since the serendipitous discovery of chlorpromazine over half a century ago [3], antipsychotics have become the cornerstone of treatment of SDDs [4]. However, current guidelines recommend that all patients should be offered cognitive behavioural therapy and family intervention in addition to antipsychotics [5]. Above and beyond *remission* or symptomatic improvement, *recovery* has become the top goal within current guidelines [6]. Achieving recovery includes outcomes such as general functioning and quality of life [7], reported to be impaired in most mental disorders [8]. While there is no single definition of recovery, the guiding principle should be that of hope, that is, the process through which patients regain a meaningful life despite suffering a serious mental illness.

Specifically, *functioning* has been defined as a ‘dynamic interaction between a person’s health condition, environmental and personal factors’, while *disability* may be considered as ‘the negative side of it’ [9]. From the *subjective* patient’s perspective, *quality of life* is defined as ‘an individual’s perception of their position in life in the context of the culture and value systems in which they live and in relation to their goals, expectations, standards and concerns’ [10]. Functioning and disability are therefore assessed by trained physicians, based on their own observations and information from others, i.e., they are *objective* measures. However, quality of life, which is *subjective*, is self-reported. Hence, a better understanding of the determinants of functional outcomes in SSD becomes a matter of major clinical relevance, which may lead to the identification of potential treatment targets. In this regard, previous research showed neurocognitive deficits [8,11,12], positive and negative symptoms [8], insight [13] and metacognition [14] to influence global functioning in SSD. 

Over the last few years there has been growing research interest in metacognition, defined as ‘knowledge and cognition about cognitive phenomena’ [15] or ‘the ability to think of one’s own and others’ thinking’ [16], which is known to be impaired in SSD patients [17,18]. Thus, Metacognitive Training [19], which addresses cognitive and metacognitive skills, was demonstrated to improve positive symptoms [20,21,22,23] and insight [24,25] in SSD.

Specifically, two core metacognitive domains have attracted attention from psychosis researchers. First, mentalizing or ‘Theory of Mind’ (ToM), defined as ‘the ability to attribute mental states—beliefs, intents, desires, emotions and knowledge—to oneself and to others’ [26] was reported to be impaired in patients with psychosis from the first presentation [27]. Second, Cognitive Insight, which includes a person’s ability to evaluate and correct his/her own distorted beliefs and misinterpretations (self-reflectiveness) and the tendency to have overconfidence in one’s conclusions (self-certainty) [17], was found to be impaired in SSD patients compared with the general population [17]. In addition, poor decision-making skills have been associated with psychosis. Decision making can be defined as ‘the ability to choose between two or more options’. However, further theoretical debate and empirical research are needed to clarify such a complex concept and its clinical correlates [28]. In particular, some physical underpinnings of decision making have been reported in healthy university students, such as heart rate variability [29]. More specifically, so-called ‘Jumping to conclusions’ (JTC) cognitive bias, i.e., drawing a conclusion based on insufficient evidence, has been reported to be greater in patients with psychotic disorders than in healthy controls [30,31,32]. Therefore, ToM, Cognitive Insight and JTC may affect functional outcome, although their specific influence remains to be established. While better cognitive and metacognitive function may be expected to go along with better outcomes, some measures of insight have shown the opposite tendency in relation to subjective quality of life [33]. This is perhaps explained through heightened awareness of the (negative) consequences of illness.

We aimed to investigate the role of clinical and cognitive insight, JTC and ToM in general functioning, disability and quality of life in a sample of outpatients with SSD, whilst adjusting for demographic, clinical, neurocognitive and psychopathological variables. We hypothesised: (i) that general functioning and disability, both of which are objective measures of functional outcome, will be determined by good premorbid adjustment, mild symptomatic severity and good neuro- and metacognitive performance (i.e., higher functioning levels and lower disability levels) [11,34]; and (ii) that quality of life, which is subjective, will correlate with insight and depression, i.e., greater insight and lower mood linked with worse quality of life [35].

## 2. Materials and Methods

### 2.1. Sample

Baseline data from a randomised controlled trial (RCT) of metacognitive training [36], which was carried out at the Hospital Universitario Fundación Jiménez Díaz (Madrid, Spain) from June 2019 to September 2020, were used for this study. Briefly, those outpatients (age 18–64 years) diagnosed with SSD, based on the Mini International Neuropsychiatric Interview, 5th Edition (MINI) [37], were approached and invited to participate in the RCT. Recruitment began on the 10 June 2020 and had to be stopped on the 11 March 2020 due to the COVID-19 outbreak in Spain. Exclusion criteria were: (i) IQ ≤ 70, which was assessed with the short form of the Wechsler Adult Intelligence Scale (WAIS)-IV [38], (ii) organic psychosis; (iii) having received a metacognitive intervention within the previous year; (iv) poor Spanish fluency; (v) lack of cooperativeness for participating in the intervention groups detailed below, both of which were judged by the treating consultant psychiatrist or psychologist and checked by the principal investigator of the project during the screening/informed consent process. Those who agreed to enroll in the RCT gave written informed consent. This RCT obtained ethical approval from the Local Research Ethics Committee (EC044-19_FJD-HRJC) and is registered at ClinicalTrials.gov (NCT04104347).

### 2.2. Variables

#### 2.2.1. Outcome Measures

General functioning, disability and QoL were the main outcome measures of this study and assessed with the Global Assessment of Functioning (GAF) [39], the World Health Organization Disability Assessment Schedule (WHODAS) [40] and the Satisfaction Life Domains Scale (SLDS) [41], respectively.

*General Functioning* was assessed with the Global Assessment of Functioning (GAF) [39], which is a rater-based 100-point Likert scale ranging from 1 (very poor functioning) to 100 (very good functioning) according to patient’s observed functionality over the last week. Of note, the inter-rater reliability for research use was reported to be good to excellent, with intra-class correlations coefficients ranging from *r* = 0.81 to *r* = 0.85 [42].

The 12-item version of the World Health Organization Disability Assessment Schedule (WHODAS 2.0) [40] was used to evaluate disability. The WHODAS is an assessor-based questionnaire enquiring about general disability-related issues, which participants answer within a 0 (lack of disability) to 4 (disabled) Likert scale. Scores for individual items can be summed up to create total scores. Higher scores indicate more severe disability, i.e., poorer functioning. Of note, a recent study from our group showed the WHODAS 2.0 to adequately measure disability aspects in both severe and common mental disorders [43]. Moreover, in an independent sample of outpatients with psychotic disorders the WHODAS 2.0 total score showed good reliability (Cronbach’s α = 0.89) [44].

QoL was measured with the Spanish version of the Satisfaction Life Domains Scale (SLDS) [41], which is a 15-item self-reported scale to measure patients’ satisfaction with their own lives. Each question is scored within a 1 to 7 Likert Scale and individual scores can be summed up to create total scores—the higher the score, the better the QoL. Specifically, in the aforementioned validation study of the Spanish version of the SLDS with a sample of patients with schizophrenia, the instrument was found to be valid and reliable, with intraclass correlation coefficients oscillating between 0.51 and 0.83 [41].

As detailed above, while the SLDS was self-rated, the same researcher (JDLM) administered the GAF and the WHODAS 2.0 to the whole sample, thus avoiding inter-rater reliability issues.

#### 2.2.2. Independent Variables

Three demographic variables were recorded: age, gender (male/female), education level (primary/above primary), marital status (unmarried/married), employment status (unemployed/employed) and living status (alone/with someone else). We also collected data on ICD-10 diagnosis based on the MINI [37], previous suicidal behaviour (present/absent), illness duration (≤5 years vs. >5 years) and number of previous psychiatric admissions.

Premorbid adjustment was retrospectively rated with the Premorbid Adjustment Scale (PAS) [45]. Specifically, the PAS provides scores on the level of adjustment over (i) childhood (to age 11), (ii) early adolescence (age 11–15) and (iii) late adolescence (age 15–17) by inquiring about sociability and social withdrawal, peer relationships, scholastic performance, adaptation to school and ability to form socio-sexual relationships.

The Spanish version [46] of the Positive and Negative Syndrome Scale (PANSS) [44] was used to rate severity of five symptomatic dimensions, namely positive, negative, disorganization, mania and depression, based on a previous consensus of PANSS factor analysis studies [47].

The vocabulary subtest of the Wechsler Adult Intelligence Scale (WAIS)-IV [38] estimated participants’ IQ; and the Trail Making Test (TMT) [48]—time to complete task B (in seconds) minus time to complete task A—provided a brief measure of executive function, whilst adjusting for processing speed [49].

Insight (i.e., clinical insight) was measured with the Spanish version [50] of the Schedule for Assessment of Insight (SAI-E) [51]. The scale, which takes the form of an observer-rated semi-structure interview, measures three insight dimensions—illness recognition, symptoms relabelling and treatment compliance—based on David’s model of insight [52], which can be summed up to provide a global insight score. Higher scores represent better insight.

*Jumping to Conclusions (JTC)* cognitive bias was measured with a computerised version of the *Beads Task* [53]. On the basis of probability (in task 1 the probability is 85:15, while in task 2 the probability is 60:40), the individual must decide the jar to which the extracted bead belongs, that is, for each task only one trial was permitted to avoid learning. *JTC* was rated if a decision was made after extracting one or two beads. This dichotomic measure of JTC as present/absent based on the ‘two or less draw to decision threshold’ was found to be most reliably associated with delusions [30] and widely used in previous early onset psychosis studies [24,32,54].

*Cognitive insight* was assessed with the *Beck Cognitive Insight Scale* (BCIS) [17], Spanish version [55]. The BCIS takes the form of a 15-item self-rated scale which yields two factors, namely self-reflectiveness (9 items) and self-certainty (6 items). For each item, scores ranged from 0 (“do not agree at all”) to 4 (“agree completely”). An overall measure of cognitive insight—Composite Index—can be calculated by subtracting self-certainty from self-reflectiveness. High self-reflectiveness, low self-certainty and high Composite Index scores are meant to indicate ‘good’ cognitive insight. Internal consistency was found to be acceptable (α = 0.60–0.68) [24]. *Mentalizing* or *Theory of Mind (ToM)* was measured with the *Hinting Task* [56], Spanish version [55], which was found to have good internal consistency (α = 0.64) [24]. In addition, we used the *Emotions Recognition Test Faces* activity (ERTF) [57] to further assess ToM.

In particular, we administered the short version of the Hinting Task, which consists of two brief stories, to avoid biases related to potential cognitive issues. The stories have two characters and, at the end of each story, one of the characters drops a fairly clear hint. The subject is asked about the meaning of this character. If the response is correct, the score is 2 and no further questions about the story will be asked. If not, further information can be provided to make the hint clearer. If the response is correct on the second occasion (that is, after being given a clearer hint), the score is 1. Otherwise, the score would be 0. Hence, the sum of both stories scores yielded total scores ranging from 0 (very poor ToM performance) to 4 (very good ToM performance).

The ERTF is composed of 20 different photographs showing people’s emotions, which participants are asked to recognise between two given options (e.g., “angry” or “sad”). Right answers score 1, while wrong answers score 0. Individual items scores can thus be summed up to create total scores. Higher scores on each scale indicate better ToM performance.

#### 2.2.3. Statistics

First, we conducted bivariate correlations between GAF, WHODAS and SLDS total scores, and all the potential confounders to select those variables which entered the hierarchical regression analyses, thus no correction for multiple testing was applied. Second, those correlated variables at a significant level (*p* < 0.10) were entered into three hierarchical multiple regressions on the three functional outcome measures, namely GAF, WHODAS and SLDS total scores. Those potential confounders which correlated (at *p* < 0.10) with each of the outcome measures were entered into the models as prior blocks (enter method). As a result, no correction for multiple testing was needed.

The percentage of the variance on GAF, WHODAS and SLDS scores explained by each model, and the contribution of each independent block to the model was investigated (Nagelkerke R^2^ change), which was considered *clinically meaningful* if R^2^ change ≥ 0.08 or *p* < 0.05 (two-tailed) [58]. The Statistical Package for Social Science version 25.0 (SPSS Inc., Chicago, IL, USA) was used to perform the analyses.

As noted above, this study was part of a larger randomised controlled trial (RCT) on insight (measured with the SAI-E) as primary outcome, on which the estimation of the sample size was based. In particular, for the RCT primary outcome (i.e., SAI-E total score) a total sample size of *n* = 102 participants at the end of the trial would be needed to detect a medium effect size (*d* = 0.50, α = 5%, 1-β = 80%). This said, as of 11 March 2020, when recruitment had to be stopped due to the COVID-19 outbreak in Spain, *n* = 351 individuals were found eligible. However, only *n* = 77 of them (21.9%) agreed to take part in the trial and were assessed at baseline (from which this study data came), which was mainly due to not consenting (*n* = 243, 69.2%).

## 3. Results

### 3.1. Sample Characteristics

The socio-demographic and clinical characteristics of the sample (*n* = 77), including psychopathological, insight-related, neurocognitive, metacognitive and functioning variables, are shown in Table 1.

### 3.2. Bivariate Analyses

Table 2, below, shows the relationship of the three outcome variables of interest—general functioning, disability and QoL—with the analysed binary independent variables. Employment status was the only variable significantly associated with the GAF score (unemployed vs. employed: 59.9 ± 6.9 vs. 66.8 ± 5.7, *p* < 0.001), although the duration of illness (≤5 years vs. >5 years: 66.0 ± 8.9 vs. 61.3 ± 7.2, *p* < 0.094) and JTC (Present vs. Absent: 60.40 ± 6.7 vs. 63.5 ± 8.2, *p* < 0.076) were associated with the GAF score at the borderline level. Those with previous suicidal behaviour were found to have significantly higher WHODAS scores than those without such antecedents (18.4 ± 9.9 vs. 12.9 ± 10.3, respectively, *p* = 0.02). Further bivariate relationships between general functioning, disability and QoL, and other binary variables are shown in Table 2.

Table 3, below, shows, the correlations between the three main outcomes of the study, namely general functioning (GAF total score), disability (WHODAS total score) and QoL (SLDS total score), and tested continuous variables.

Age (*r* = −0.31, *p* = 0.005), the late adolescence PAS score (*r* = −0.22, *p* = 0.060), PANSS-Negative (*r* = −0.52, *p* < 0.001), PANSS-Disorganization (*r* = −0.43, *p* < 0.001), and TMT B-A (*r* = −0.27, *p* = 0.021), were found to have a negative association with the GAF, while IQ (*r* = 0.22, *p* = 0.054), illness recognition (*r* = 0.34, *p* = 0.002), BCIS-SR (*r* = 0.21, *p* = 0.075), the Hinting Task score (*r* = 0.25, *p* = 0.028) and ERTF (*r* = 0.34, *p* = 0.002) had a positive correlation with the GAF.

PANSS-Negative (*r* = 0.32, *p* = 0.005) and PANSS-Depression (*r* = 0.35, *p* = 0.002) positively correlated with the WHODAS total score.

PANSS-Depression (*r* = −0.46, *p* < 0.001), illness recognition (*r* = −0.28, *p* = 0.015) and BCIS-SR (*r* = −0.23, *p* = 0.051) showed a negative correlation with the SLDS score.

Further associations of functioning, disability and quality of life scores with sociodemographic, clinical, neurocognitive and metacognitive variables are presented in Table 3.

### 3.3. Multivariable Regression Models

Table 4, below, presents the hierarchical multivariable regression models on the three outcome measures, namely general functioning-GAF, disability-WHODAS and quality of life-SLDS.

#### 3.3.1. General Functioning—GAF

Age, employment status, duration of illness, a late adolescence PAS score, PANSS negative and disorganization symptoms, illness recognition, BCIS-SR, JTC and ToM variables (Hinting Task and ERTF) were added to the model, although only age and employment status (R^2^ change = 0.211; *p* = 0.001), a late adolescence PAS score (R^2^ change = 0.049; *p* = 0.050), PANSS—disorganization and negative factors—(R^2^ change = 0.087; *p* = 0.025) and ToM (R^2^ change = 0.066, *p* = 0.053), remained significant. This model explained 41.3% of the variance on the total GAF score.

#### 3.3.2. Disability—WHODAS

Previous suicidal behaviour (R^2^ change = 0.068; *p* = 0.023) and PANSS—negative and depressive factors—(R^2^ change = 0.167; *p* = 0.001) contributed to disability, accounting for 23.5% of the variance on the total WHODAS score.

#### 3.3.3. Quality of Life—SLDS

Previous suicidal behaviour (R^2^ change = 0.070, *p* = 0.026), PANSS-Depression (R^2^ change = 0.157; *p* < 0.001) and illness recognition (R^2^ change = 0.046, *p* = 0.044) were significantly associated with QoL, explaining 27.3% of the variance on total SLDS score.

## 4. Discussion

### 4.1. Principal Findings

We conducted this cross-sectional study based on data from an RCT with SSD outpatients, aimed to investigate the contribution of clinical and cognitive insight, Jumping to Conclusions (JTC) and Theory of Mind (ToM) to three measures of functional outcome, namely general functioning, disability and quality of life (QoL), whilst adjusting for a set of demographic, clinical, neurocognitive and psychopathological variables. In light of our results two main conclusions can be drawn.

Consistent with our first hypothesis, we found that an overall measure of general functioning, such as the GAF score, was predicted by age (being young), good premorbid adjustment, particularly in late adolescence, less severe disorganization and negative symptoms and better ToM performance. However, only previous suicidal behaviour, negative and depressive symptoms predicted disability, which partially conflicted with hypothesis i. Thus, suicide attempters were found to be more disabled than non-attempters and more severe negative and disorganization symptoms were associated with increased disability.

Our second hypothesis was partially supported by results which revealed previous suicidal behaviour, depressive symptoms’ severity, and insight into having a mental illness to predict QoL. Specifically, non-suicide attempters had a better QoL than attempters, and more severe depressive symptoms and greater insight were linked with a worse QoL.

### 4.2. General Functioning

As noted above, general functioning can be considered an *objective* measure of overall psychosocial functioning of patients. Although achieving recovery has become paramount, according to current guidelines [6], what underlies functioning in SSD remains far from clear.

Specifically, neurocognitive deficits commonly observed in SSD patients were related to objective measures of QoL—but not with subjective-QoL—[11] and general functioning [8,11,12]. Furthermore, executive training proved effective in improving neurocognition and functioning [59]. Above and beyond neurocognition, social cognition [60] and metacognition [14] were proposed to mediate the relationship between neurocognition and functioning. In keeping with this, this study’s results found illness recognition and self-reflectiveness to predict general functioning (assessed with the GAF) at a borderline level (*p* = 0.072), which may have reached significance in a larger sample.

Additionally, we replicated the role of negative symptoms in objective-QoL, that is, a proxy for functioning [61]. Somehow in line with this, positive symptoms’ severity was related to poorer general functioning—as measured with the GAF—and QoL in a sample of patients with the first-episode of psychosis (FEP) [62], thus linking functioning with QoL. This finding was consistent with a systematic review of 42 studies (*n* = 8250 participants) which found functioning to be directly related with QoL, although this association was suggested to be somewhat mediated by the QoL assessment instrument [63].

In summary, age (i.e., being young), being employed, better premorbid adjustment, less severe symptoms, better neuro- and metacognitive performance, and higher cognitive and clinical insight levels were associated with greater functioning in our sample of SSD patients based on the bivariate analyses. It should be noted, however, that among these determinants of general functioning, only symptom severity and ToM (the latter at a borderline level in the hierarchical multivariable regression models) can be modified through intervention. In other words, achieving recovery in SSD appears to require interventions targeting ToM deficits, such as so-called metacognitive interventions, although future trials are needed to establish this.

### 4.3. Disability

Disability was explained as the negative result of the interaction between a person’s health condition, environmental factors and personal factors. However, only a few studies tested potential predictors of disability in SSD, which is of concern, particularly taking into account the high levels of disability among SSD patients [64]. Specifically, we postulated that those with poorer premorbid adjustment, more severe psychotic symptoms and worse neuro- and metacognitive functioning would have higher disability levels. However, results supported our expectations partially. In particular, we found negative and depressive symptoms’ severity to predict disability, which was in line with a recent first-episode psychosis (FEP) study in which disability was measured with the WHODAS [65]. Somewhat unexpected, previous suicidal behaviour remained associated with disability in the multivariable regression model, thus replicating findings from a previous 1-year follow-up FEP cohort [66], that is, suicide attempters were found to have higher disability levels than non-attempters.

However, other contributors to disability were reported by previous studies with SSD samples, such as gender (female) and education level [67], sociopathic and schizotypy personality traits [66], and employment status [68]. Therefore, it seems that several factors appear to contribute to disability in SSD, although only negative and depressive symptoms and previous suicidal behaviour predicted disability in our sample of SSD patients. Certainly, lack of treatments for negative symptoms in SSD [69,70] prevents SSD individuals from recovery, which requires future RCTs. On the other hand, this finding highlights the importance of assessing depressive symptoms in SSD patients, including implementation of evidence-based treatments [71].

### 4.4. Quality of Life

Quality of Life (QoL), which is a subjective measure of one’s psychosocial functioning, has not been sufficiently investigated in SSD, which is probably due to a lack of consensus regarding its assessment. This noted, improvement in QoL is likely to be the most important goal for patients since it involves their own perception of the disease.

In particular, several predictors of QoL have been established, including education level, marital status and urbanicity, according to a recent cross-sectional study with a sample of *n* = 351 patients with schizophrenia [12] which, however, failed to link gender with QoL. Other previous works failed to relate QoL with sociodemographic data [8,72], which was in full agreement with our results.

Of relevance, the influence of symptoms on QoL has been subject to much previous research work. Thus, negative [8,12,72,73,74], positive [8,12,75], general psychopathology [12,76], depressive [61], depressive/anxiety symptoms [77] and personality traits [78] were found to be associated with QoL. Hence, symptoms remission could be thought to contribute to better QoL, consistent with a study which showed prolonged remission to improve QoL [79].

Our results replicated the role of depressive symptoms in QoL [61,62,77,80], in line with hypothesis ii. Future trials targeting depressive symptoms are therefore needed to improve QoL in patients with SSD. Insight was also found to influence QoL, which was in line with some previous studies [77]. However, whether depressive symptoms may mediate the effect of illness awareness on QoL remains unclear [35].

Interestingly, illness recognition, which can be considered as the classic insight dimension, negatively correlated with QoL. Although this finding went against the notion that illness recognition improves functioning, the subjective nature of QoL compared with general functioning (which is an objective measure of outcome), may explain this. Nevertheless, insight was associated with depressive symptoms [77], which may mediate the effect of the awareness of illness on QoL [33] through thoughts on the negative consequences of illness. In keeping this, suicide attempters were found to have worse QoL than non-attempters, which may have been due to the mediating effect of depression and/or insight on such an association. This noted, future research is needed to clarify these complex associations.

Most importantly, no psychosocial intervention has been demonstrated to improve QoL in SSD to date [81], although future trials testing novel therapies, such as metacognitive training [19], may shed some light on this.

### 4.5. Strengths and Limitations

To the best of our knowledge, this is the first study in testing the contribution of clinical and cognitive insight, JTC and ToM to three measures of global functional outcome, such as general functioning, disability and QoL in an unselected sample of SSD patients. In addition, we controlled for a wide range of demographic, clinical, neurocognitive and psychopathological variables. While this study, therefore, makes a novel contribution to the field, future replication studies are needed.

However, we acknowledge that this study has some limitations. First of all, the sample was comprised of outpatients living in an inner-city area in Madrid (Spain), thus these results might not be applicable to inpatients and to those living in rural areas. Second, other non-evaluated variables may have influenced the outcome measures, such as life style, physical activity and self-esteem [82]. Specifically, self-esteem may mitigate the detrimental effect of internalized stigma on QoL [83]. Regretfully, we did not collect data on self-esteem. Third, the cross-sectional design of the study did not allow us to capture variables’ changes over time. Hence, much caution is needed when drawing causality conclusions. Finally, the relatively small sample size may have lacked sufficient power to properly study some associations.

### 4.6. Clinical Implications and Directions for Future Research

Functional outcome in SSD has become the main treatment targets in current guidelines [6]. Above and beyond symptom remission, patients intend to regain a *normal* life within their communities, that is, *living without disability* (despite suffering a mental illness) a life they feel to be satisfying and of good quality. Hence, achieving recovery in SSD requires a better understanding of the underpinnings of functional outcomes, which led us to carry out this study.

In particular, based on our results, symptomatic remission and improving ToM deficits emerged as the main modifiable factors leading to recovery since age, employment status, premorbid adjustment and previous suicidal behaviour are not prone to intervention. The question, therefore, arises. How can we achieve this? More specifically, can we improve ToM performance and insight in SSD? A recent systematic review and meta-analysis from our group found metacognitive training (MCT), which addresses metacognitive abilities, to improve insight [25] in SSD, which might direct future research in this area. In particular, whether MCT also improves the functional outcome in SSD remains unclear, which requires future RCTs.

## Figures and Tables

**Table 1 behavsci-12-00028-t001:** Sample characteristics (*n* = 77).

Socio-Demographic Variables	
Age (years)	47.69 ± 9.76
Gender (males)	41 (53.2%)
Education level (primary)	13 (16.9%)
Marital status (unmarried)	61 (79.2%)
Employment status (unemployed)	56 (72.7%)
Living status (alone)	8 (10.4%)
Premorbid Adjustment (PAS)	
Childhood	5.80 ± 3.79
Early adolescence	7.64 ± 4.64
Late adolescence	7.69 ± 4.90
Clinical variables	
Diagnosis	
Schizophrenia	48 (62.3%)
Other psychoses	29 (37.7%)
Previous suicidal behaviour (present)	31 (40.3%)
Duration of illness (>5 years)	69 (89.6%)
Number of previous admissions	3.46 ± 3.99
Psychopathology (PANSS)	
Positive	8.44 ± 3.67
Negative	14.91 ± 5.89
Disorganisation	6.05 ± 2.61
Depression	6.25 ± 1.86
Mania	6.94 ± 2.70
Insight (SAI-E)	
Total Insight	15.55 ± 2.29
Illness Recognition	5.36 ± 2.68
Symptoms relabelling	5.87 ± 2.81
Treatment Compliance	4.31 ± 1.57
Neurocognition	
IQ	104.61 ± 11.72
TMT B-A	68.91 ± 43.65
Metacognition	
JTC_85:15	42 (56.0%)
JTC_60:40	38 (51.4%)
BCIS-SR	15.43 ± 5.11
BCIS-SC	7.67 ± 3.42
BCIS-CI	7.74 ± 6.66
Hinting Task	2.25 ± 1.33
ERTF	16.86 ± 2.16
Functioning	
General Functioning (GAF)	61.81 ± 7.48
Disability (WHODAS)	15.40 ± 10.14
Quality of Life (SLDS)	80.14 ± 11.46

PAS = Premorbid Adjustment Scale, PANSS = The Positive And Negative Syndrome Scale, SAI-E = The Schedule for Assessment of Insight—Expanded version, IQ = Intelligence Quotient, TMT B-A = Trail Making Test (time B-A), JTC = Jumping to Conclusions, BCIS-SR = Beck Cognitive Insight Scale—Self Reflectiveness, BCIS-SC = Beck Cognitive Insight Scale—Self Certainty, BCIS-CI = Beck Cognitive Insight Scale—Composite Index, ERTF = Emotions Recognition Test Faces, GAF = Global Assessment of Functioning, WHODAS = World Health Organization Disability Assessment Scale, SLDS = Satisfaction with Life Domains Scale.

**Table 2 behavsci-12-00028-t002:** Bivariate analyses: relationship between binary variables and functional outcome measures.

	General Functioning (GAF)			Disability (WHODAS)			Quality of Life (SLDS)		
	Mean (SD)	Statistic	*p* Value	Mean (SD)	Statistic	*p* Value	Mean (SD)	Statistic	*p* Value
*Gender*									
Males	61.83 (8.20)			13.49 (11.65)			80.97 (11.44)		
Females	61.78 (6.69)	*t_75_* = 0.03	0.98	17.08 (8.48)	*t_74_* = −1.52	0.13	79.25 (11.58)	*t_70_* = 0.63	0.53
*Education level*									
Primary	60.23 (9.18)			12.23 (5.99)			83.73 (11.90)		
Secondary (or above)	62.13 (7.13)	*t_75_* = −0.83	0.41	15.75 (11.04)	*t_31.70_* = −1.62	0.11	79.49 (11.36)	*t_70_* = 1.13	0.26
*Marital status*									
Unmarried	61.57 (7.70)			15.26 (9.97)			80.28 (10.55)		
Married	62.69 (6.74)	*t_75_* = −0.053	0.60	14.67 (12.40)	*t_74_* = 0.20	0.84	79.60 (14.84)	*t_70_* = 0.20	0.84
*Living status*									
Alone	61.13 (9.30)			20.50 (9.43)			76.00 (10.02)		
Not alone	61.88 (7.32)	*t_75_* = −0.27	0.79	14.51 (10.39)	*t_74_* = 1.55	0.12	80.58 (11.58)	*t_70_ *= −1.01	0.32
*Employment status*									
Unemployed	59.93 (6.91)			15.67 (10.06)			80.39 (11.62)		
Employed	66.81 (5.74)	*t_75_* = −3.92	<0.001	13.76 (11.40)	*t_74_* = 0.71	0.48	79.52 (11.30)	*t_70_* = 0.29	0.77
*Diagnosis*									
Schizophrenia	61.73 (7.45)			15.77 (10.67)			80.06 (11.63)		
Other psychoses	61.93 (7.68)	*t_75_* = −0.11	0.91	14.07 (10.02)	*t_74_* = 0.68	0.50	80.27 (11.38)	*t_70_* = −0.07	0.94
*Previous SB*									
Present	62.35 (7.56)			18.39 (9.87)			77.24 (12.55)		
Absent	61.43 (7.49)	*t_75_* = 0.53	0.60	12.91 (10.27)	*t_74_* = 2.32	0.02	82.09 (10.35)	*t_70_* = −1.79	0.078
*Duration of illness*									
≤5 years	66.00 (8.91)			14.50 (11.40)			74.00 (9.43)		
>5 years	61.32 (7.22)	*t_75_* = 1.69	0.094	15.22 (10.37)	*t_74_* = 0.18	0.85	80.80 (11.52)	*t_70_* = 1.50	0.14
*JTC*									
Present	60.40 (6.75)			15.12 (9.30)			81.35 (10.73)		
Absent	63.52 (8.21)	*t_73_* = −1.80	0.076	15.50 (11.96)	*t_72_* = −0.015	0.88	78.21 (12.27)	*t_68_* = 1.14	0.26

GAF = Global Assessment of Functioning, WHODAS = World Health Organization Disability Assessment Scale, SLDS = Satisfaction with Life Domains Scale, SB = Suicidal Behaviour, JTC = Jumping to Conclusions.

**Table 3 behavsci-12-00028-t003:** Bivariate correlations of continuous variables with functional outcome measures.

	General Functioning (GAF)		Disability (WHODAS)		Quality of Life SLDS	
	*r*	*p*	*r*	*p*	*r*	*p*
Age	−0.31	0.005	0.17	0.13	0.069	0.57
PAS Child	−0.12	0.29	0.14	0.21	−0.12	0.31
PAS early	−0.11	0.32	−0.02	0.89	0.00	>0.99
PAS late	−0.22	0.060	0.03	0.81	−0.04	0.74
PANSS-Positive	−0.00	0.97	0.14	0.24	−0.18	0.13
PANSS-Negative	−0.52	<0.001	0.32	0.005	−0.09	0.47
PANSS-Disorganization	−0.43	<0.001	0.10	0.39	0.09	0.44
PANSS-Mania	−0.00	0.98	−0.05	0.64	0.06	0.60
PANSS-Depression	−0.24	0.83	0.35	0.002	−0.46	<0.001
IQ	0.22	0.054	0.08	0.47	−0.14	0.24
TMT_B-A	−0.27	0.021	0.03	0.81	0.13	0.27
Recognition	0.34	0.002	0.13	0.26	−0.28	0.015
Relabelling	−0.02	0.87	−0.02	0.89	0.01	0.90
Compliance	−0.04	0.70	0.13	0.27	0.01	0.92
INSIGHT	0.15	0.19	0.10	0.41	−0.13	0.26
BCIS-SR	0.21	0.075	0.06	0.59	−0.23	0.051
BCIS-SC	−0.04	0.73	−0.05	0.65	−0.05	0.69
BCIS-CI	0.18	0.13	0.07	0.54	−0.17	0.17
Hinting Task	0.25	0.028	−0.09	0.45	0.07	0.59
ERTF	0.34	0.002	−0.03	0.80	−0.13	0.27

GAF = Global Assessment of Functioning, WHODAS = World Health Organization Disability Assessment Scale, SLDS = Satisfaction with Life Domains Scale, PAS = Premorbid Adjustment Scale, PANSS = The Positive And Negative Syndrome Scale, IQ = Intelligence Quotient, TMT B-A = Trail Making Test (time B-A), BCIS-SR = Beck Cognitive Insight Scale—Self Reflectiveness, BCIS-SC = Beck Cognitive Insight Scale—Self Certainty, BCIS-CI = Beck Cognitive Insight Scale—Composite Index, ERTF = Emotions Recognition Test Faces.

**Table 4 behavsci-12-00028-t004:** Multivariable linear regression models on general functioning, disability and quality of life.

	General Functioning (GAF)			Disability (WHODAS)			Quality of Life (SLDS)		
	R^2^ Change	F Change	*p*	R^2^ Change	F Change	*p*	R^2^ Change	F Change	*p*
Block 1—Demographic variables: Age, employment status	0.211	7.884	0.001						
Block 2—Clinical variables: Duration of illness, previous SB	0.043 ^a^	3.341	0.073	0.068 ^b^	5.383	0.023	0.070 ^b^	5.155	0.026
Block 3—Premorbid adjustment	0.049	4.027	0.050						
Late PAS score									
Block 4—PANSS: See below.	0.087 ^c^	3.941	0.025	0.167 ^d^	7.867	0.001	0.157 ^e^	13.610	<0.001
Block 5—Neurocognition: IQ (WAIS vocabulary), TMT B-A	0.007	0.311	0.734						
Block 6—Clinical Insight:	0.023	2.109	0.161				0.046	4.216	0.044
Recognition									
Block 7—Cognitive Insight	0.001	0.097	0.757				0.001	0.069	0.793
SR									
Block 8—JTC	0.001	0.126	0.724						
JTC									
Block 9—Theory of Mind	0.066	3.120	0.053						
Hinting Task, ERTF									
	41.3%			23.5%			27.3%		

GAF = Global Assessment of Functioning, WHODAS = World Health Organization Disability Assessment Scale, SLDS = Satisfaction with Life Domains Scale, SB = Suicidal behaviour, PAS = Premorbid Adjustment Scale, PANSS = The Positive and Negative Syndrome Scale, IQ = Intelligence Quotient, WAIS = Wechsler Adult Intelligence Scale, TMT B-A = Trail Making Test (time B-A), JTC = Jumping to Conclusions, ERTF = Emotions Recognition Test Faces. ^a^ Duration of illness, ^b^ previous suicidal behaviour, ^c^ negative + disorganisation, ^d^ negative + depression, ^e^ depression.

## Data Availability

Data supporting these findings are available upon reasonable request to the authors, the provided dataset access policy is complied with.

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
