# Peer review of "Investigating the Role of Insight, Decision-Making and Mentalizing in Functional Outcome in Schizophrenia: A Cross-Sectional Study"

_behavsci, 2022, doi:10.3390/bs12020028_

Round 1
Reviewer 1 Report
The manuscript is well done.
I just suggest to improve introduction with some more accurate information about the DM.
Morelli, M., Casagrande, M., & Forte, G. (2021). Decision Making: a Theoretical Review. Integrative Psychological and Behavioral Science, 1-21.
Forte, G., Morelli, M., & Casagrande, M. (2021). Heart Rate Variability and Decision-Making: Autonomic Responses in Making Decisions. Brain Sciences, 11(2), 243.
Author Response
The reviewer congratulated the authors on this work as follows: ‘The manuscript is well done’.
In order to improve the introduction the reviewer suggested providing further information on decision-making, including two cites (Morelli et al., 2021; Forte et al., 2021).
Accordingly, we have amended the introduction section of the revised manuscript (page 2, sixth paragraph, lines 81-86) as follows:
In addition, poor decision-making skills have been associated with psychosis. Decision-making can be defined as ‘the ability to choose between two or more options’, although further theoretical debate and empirical research are needed to clarify such a complex concept and its clinical correlates (Morelli et al., 2021). In particular, some physical underpinnings of decision-making have been reported in healthy university students, such as heart rate variability (Forte et al, 2021). More specifically, so-called ‘Jumping to conclusions’ (JTC) cognitive bias, i.e., drawing a conclusion based on insufficient evidence, has been reported to be greater in patients with psychotic disorders than in healthy controls [29–31].
Reviewer 2 Report
It is my pleasure to review this well-written manuscript addressing predictors of functioning outcomes in patients with schizophrenia. I suggest some improvements:
- It will be great if the authors can briefly mention study aims in the abstract.
- The readability will be advanced if the sentences are more simple and succinct in the introduction section. For example, on page one, the sentence of "as a result....in most mental disorders (line 44-47)" is very long and difficult to follow.
- Please provide more information, such as reliability/validity, of all instruments. It is also important to know if these instruments were used in similar population previously.
- While GAF, WHODAS are subjectively rated, it is not clear who determined the ratings.
- Please provide more information regarding sampling: how many potential participants were approached and how many agreed to participate? how do you justify the sample size?
Author Response
First, the reviewer referred to the manuscript as ‘well-written’, although he/she suggested some improvements.
First, the reviewer asked the authors to briefly mention the study aims in the abstract.
Based on this point, we have amended the first sentence of the abstract in the revised manuscript as follows:
Background: Recovery has become a priority in schizophrenia spectrum disorders (SSD). This study aimed to investigate predictors of objective -general functioning and disability- and subjective -quality of life (QoL)- measures of functional outcomes in SSD.
Second, the reviewer suggested improving the readability of introduction section by shortening some sentences.
Accordingly, we have reviewed this section very carefully and in the revised manuscript we have used much shorter sentences.
Third, the reviewer recommended the authors to provide reliability/validity information on the instruments used in this study.
Accordingly, we have added this information to the methods section of the revised manuscript. For the sake of the article, we have focused on the outcomes measures, that is, GAF, WHODAS 2.0 and SLDS:
GAF (lines 131-133):
Of note, the inter-rater reliability for research use was reported to be good to excellent, with intra-class correlations coefficients ranging from r=0.81 to r=0.85 (Vatnaland et al., 2007).
WHODAS (lines 141-143):
Also, in an independent sample of outpatients with psychotic disorders the WHODAS 2.0 total score showed good reliability (Cronbach’s α=0.89) (Holmberg et al., 2021).
SLDS:
Specifically, in the aforementioned validation study of the SLDS Spanish version with a sample of patients with schizophrenia, the instrument was found to be valid and reliable, with intraclass correlation coefficients oscillating between 0.51 and 0.83 (Ochoa et al., 2017)
Fourth, the reviewer asked the authors to clarify who administered the GAF and WHODAS.
Accordingly, we have amended section 2.1.1 of the revised manuscript (page 4, lines 151-153) as follows:
As detailed above, while the SLDS was self-rated, the same researcher (JDLM) administered the GAF and the WHODAS 2.0 to the whole sample, thus avoiding inter-rater reliability issues.
Fifth, the reviewer requested more information on the estimation of the sample size and statistical power calculations.
In this respect, it is worth noting that this analysis was part of a larger randomised controlled trial (RCT) of metacognitive training, as detailed in the methods sections (see section 2.1.). Hence, the estimation of the sample size was based on the primary outcome of the RCT, namely insight as measured with the SAI-E. Nevertheless, we have clarified this in the revised manuscript (section 2.1.3, lines 228-236):
As noted above, this study was part of a larger randomised controlled trial (RCT) on insight (measured with the SAI-E) as primary outcome, on which the estimation of the sample size was based. In particular, for the RCT primary outcome (i.e., SAI-E total score) a total sample size of N=102 participants at the end of the trial would be needed to detect a medium effect size (d=0.50, α=5%, 1-β=80%). This said, as of the 11/03/2019, when recruitment had to be stopped due to the COVID-19 outbreak in Spain, N=351 individuals were found eligible. However, only n=77 of them (21.9%) agreed to take part in the trial and were assessed at baseline (from which this study data came), which was mainly due to not consenting (n=243, 69.2%).
Reviewer 3 Report
Summary:
In a sample of 77 schizophrenia spectrum outpatients, the role of several predictors, including insight, JTC bias, and ToM ability on general functioning, disability and quality of life was investigated. The study can explain large parts of the variance of all three outcome variables. Overall, the results provide a solid basis to suggest that targeted cognitive therapy addressing some of the major predictors, as may be accomplished in the scope of metacognitive training, may positively influence general functioning and quality of life in individuals with schizophrenia.
The manuscript is well-written and the reader gets a good understanding of the goals and methods of the study. The results are relevant to the research and therapy of schizophrenia spectrum disorders.
I only have some minor remarks to further improve the quality of the manuscript:
- The authors should thoroughly reread the manuscript and check for spelling mistakes (e.g., line 42 SDD instead of SSD; line 104 year when study started).
- How was Spanish fluency (line 108) assessed?
- The description of socio-demographic variables in methods and results should be expanded. E.g., in the methods section, assessment of marital or employment status is not mentioned. In the results section, the table could be clearer: Does the education level (primary) value indicate, that only 13 individuals in the sample had completed primary education?
If 48 individuals have the Diagnosis F20, what diagnosis does the rest of the sample have? - Table 4 is a bit hard to read due to the formatting, in the second column (GAF, P), there is even a leading 0 (PAS late, 0.060) in contrast to the other entries. Since at times the authors even end the numbers with a 0, I would recommend to always use 3 decimal places (fill up with 0 if necessary), or use right-alignment for the columns. Also, I assume the authors are reporting correlations r with p-values, it seems uncommon to me to write a capital P for that. In addition, the authors could indicate significant values using asterisks (*).
- The authors should check for consistency. E.g., in Table 4, at times there are colons behind the Block names, at times not; in the table description, there are at times spaces after the equal-sign, at times not; etc.
- Was there only 1 trial for each of the 2 versions of the Beads task? The authors should be more specific in the manuscript.
- It would be helpful if the authors could provide more details on the mentalizing tasks, such as what exactly the participants had to do.
- The authors should mention why they don’t perform correction for multiple testing, possibly also in the limitations section.
Author Response
The reviewer appreciated the efforts behind this work as follows: “The manuscript is well-written and the reader gets a good understanding of the goals and methods of the study. The results are relevant to the research and therapy of schizophrenia spectrum disorders”.
However, the reviewer made eight minor remarks to further improve the quality of the article.
First, the reviewer recommended the authors to carefully reread the manuscript and check for misspellings. Accordingly, we have carefully reviewed the revised manuscript and amended all the typos and misspellings.
Second, the reviewer asked for further information on the assessment of the Spanish fluency (methods section).
Although both the cooperativeness and level of Spanish were first determined by the referring consultant, during the screening/informed consent process the Principal Investigator of this project (JDLM) also double-checked this, which has been clarified further in the revised manuscript as follows (page 3, section 2.1., lines 115-119):
- iv) poor Spanish fluency; v) lack of cooperativeness for participating in the intervention groups detailed below, both of which were judged by the treating consultant psychiatrist or psychologist and checked by the principal investigator of the project during the screening/informed consent process.
Third, the reviewer asked for further clarification on the description of socio-demographic variables across the methods and results section.
Accordingly, we have amended both sections in the revised manuscript.
Fourth, the reviewer suggested rewriting table 4 due to formatting issues.
We have seriously considered this point and we have carefully reviewed all the inconsistencies. However, we respectfully disagree on reformatting the whole Table 4 which was done following ‘How to Report Statistics in Medicine: Annotated Guidelines for Authors, Editors and Reviewers (Lang & Sesic, 1997), on which the Americal College of Phisicians and the American Psychiatric Association guidelines for statistic reporting are based. In short, this is why we used P in capital letters or we changed the number of decimals depending on the level of significance.
Fifth, in keeping with the above the reviewer recommended us to check for consistency across the files.
We have appreciated this point which has allowed us to check for some typos and misspellings following a careful review of the revised manuscript, which is enclosed.
Sixth, the reviewer asked the authors if there was only one trial for each of the 2 versions of the Beads Task, which he/she found a bit unclear in the submitted manuscript.
Accordingly, we have clarified this further in the methods section of the revised manuscript (lines 181-188):
Jumping to Conclusions (JTC) cognitive bias was measured with a computerised version of the Beads Task [51]. On the basis of probability (in task 1 the probability is 85:15, while in task 2 the probability is 60:40), the individual must decide the jar to which the extracted bead belongs, that is, for each task only one trial was permitted to avoid learning. JTC was rated if a decision was made after extracting one or two beads. This dichotomic measure of JTC as present/absent based on the ‘two or less draw to decision threshold’ was found to be most reliably associated with delusions [30] and widely used in previous early onset psychosis studies [24,31,52].
Seventh, the reviewer suggested providing further details about the mentalizing tasks.
We have detailed the two mentalizing instruments as follows (lines 197-208):
Mentalizing or Theory of Mind (ToM) was measured with the Hinting Task [56], Spanish version [55], which was found to have good internal consistency (α=0.64) [24]. In addition, we used the Emotions Recognition Test Faces activity (ERTF) [57] to further assess ToM.
In particular, we administered the short version of the Hinting Task, which consists of two brief stories, to avoid biases related to potential cognitive issues. The stories have two characters and, at the end of each story, one of the characters drops a fairly clear hint. The subject is asked about the meaning of this character. If the response is correct, the score is 2 and no further questions about the story will be read. If not, further information can be provided to make the hint clearer. If the response is correct on the second occasion (that is, after being given a clearer hint), the score is 1. Otherwise, the score would be 0. Hence, the sum of both stories scores yielded total scores ranging from 0 (very poor ToM performance) to 4 (very good ToM performance).
Eighth, the reviewer asked us why we had not perform correction for multiple testing.
Truly, we did not perform correction for multiple testing since we used multivariate hierarchical regression models. We have clarified this further in the statistics section of the revised manuscript ():
First, we conducted bivariate correlations between GAF, WHODAS and SLDS scores and all the potential confounders to select those variables which entered the hierarchical regression analyses, thus no correction for multiple testing was applied. Second, those correlated variables at a significant level (P<.10) were entered into three hierarchical multiple regressions on the three functional outcome measures, namely GAF, WHODAS and SLDS scores. Those potential confounders which correlated (at P<.10) with each of the outcome measures were entered into the models as prior blocks (enter method). Therefore, we analysed the contribution of each independent variable to the outcome variable, whilst adjusting for each all the other tested contributors. Therefore, we analysed the contribution of each independent variable to the outcome variable, whilst adjusting for all the other tested contributors. As a result, no correction for multiple testing was needed.